# G-Protein-Coupled Estrogen Receptor Expression in Rat Uterine Artery Is Increased by Pregnancy and Induces Dilation in a Ca^2+^ and ERK1/2 Dependent Manner

**DOI:** 10.3390/ijms23115996

**Published:** 2022-05-26

**Authors:** Teresa Tropea, Damiano Rigiracciolo, Milena Esposito, Marcello Maggiolini, Maurizio Mandalà

**Affiliations:** 1Maternal and Fetal Health Research Centre, Division of Developmental Biology and Medicine, Faculty of Biology, Medicine and Health, University of Manchester, Manchester M13 9WL, UK; teresa.tropea@manchester.ac.uk; 2Manchester Academic Health Science Centre, Manchester University NHS Foundation Trust, St. Mary’s Hospital, Manchester M13 9WL, UK; 3Department of Pharmacy, Health and Nutritional Sciences, University of Calabria, 87036 Rende, Italy; damianorigiracciolo@yahoo.it (D.R.); marcello.maggiolini@unical.it (M.M.); 4Department of Biology, Ecology and Earth Sciences, University of Calabria, 87036 Rende, Italy; milenaesposito17@gmail.com

**Keywords:** estrogen receptors, resistance arteries, vasorelaxation, pregnancy and blood vessels

## Abstract

Increasing levels of estrogens across gestation are partly responsible for the physiological adaptations of the maternal vasculature to pregnancy. The G protein-coupled estrogen receptor (GPER) mediates acute vasorelaxing effects in the uterine vasculature, which may contribute to the regulation of uteroplacental blood flow. The aim of this study was to investigate whether GPER expression and vasorelaxation may occur following pregnancy. Elucidation of the functional signalling involved was also investigated. Radial uterine and third-order mesenteric arteries were isolated from non-pregnant (NP) and pregnant rats (P). GPER mRNA levels were determined and—concentration–response curve to the GPER-specific agonist, G1 (10^−10^–10^−6^ M), was assessed in arteries pre-constricted with phenylephrine. In uterine arteries, GPER mRNA expression was significantly increased and vasorelaxation to G1 was significantly enhanced in P compared with NP rats. Meanwhile, in mesenteric arteries, there was a similar order of magnitude in NP and P rats. Inhibition of L-type calcium channels and extracellular signal-regulated kinases 1/2 significantly reduced vasorelaxation triggered by G1 in uterine arteries. Increased GPER expression and GPER-mediated vasorelaxation are associated with the advancement of gestation in uterine arteries. The modulation of GPER is exclusive to uterine arteries, thus suggesting a physiological contribution of GPER toward the regulation of uteroplacental blood flow during pregnancy.

## 1. Introduction

Normal pregnancy is associated with physiological adaptations of the maternal vasculature and involves significant changes to both systemic and tissue-specific vascular beds [1,2]. These adaptations are widely triggered by a decrease in vascular tone and an increase in uterine blood flow, which is essential to sustaining sufficient uteroplacental perfusion required by the developing fetus throughout pregnancy [3,4]. 

Amongst the main mechanisms, vasoactive molecules released from the endothelium, such as nitric oxide (NO) [4,5,6], prostacyclin [7,8,9] and hyperpolarizing factor [10,11], play a key role in modulating pregnancy-associated vascular tone and reactivity, which may be regulated by the mobilization of intracellular calcium (Ca^2+^) [9,11] and activation of protein kinases, like extracellular signal-regulated kinases (ERK) [9]. Sex hormones, including estrogens, can profoundly influence vascular functions [12,13]. In this regard, it has been shown that natural estrogens stimulate the production of NO and other endothelium-derived factors [13,14,15,16], and L-type Ca^2+^ channels may be a target of acute estrogenic vasodilatory effects during pregnancy [17]. 

Estrogen-induced vasodilation has the beneficial role of maintaining arterial health [18,19], with a potentially different degree of vascular effects according to tissue-specific vascular beds. In particular, estrogens exhibit potent vasodilatory properties reaching their greatest effects on the reproductive tissues, where systemic infusion of these hormones increases uterine blood flow up to 10-fold in non-pregnant ewes [20]. 

Previous studies have established that estrogens exert a functional role in the vascular adaptation to pregnancy and regulation of uterine blood flow [21,22] through binding to the estrogen receptor (ER), Erα, and ERβ [16,23,24]. In addition to the action elicited through these receptors, estrogens activate the G protein-coupled estrogen receptor (GPER) signalling leading to acute vascular effects [25,26]. Notably, we have demonstrated that GPER is expressed in rat uterine arteries, and its activation by the specific agonist G1 induces endothelium-dependent vasorelaxation through the NO-cyclic guanosine monophosphate (cGMP) pathway [27]. Moreover, previous evidence has shown that -plasma levels of estrogens increase progressively in human pregnancy [28] as well as in pregnant rats [29].

However, it is currently unknown whether functional GPER expression changes during pregnancy in the uterine and systemic circulation. Here, we ascertained that pregnancy modulates GPER expression and GPER-mediated vasorelaxation only in uterine arteries. In addition, we investigated mechanisms underlying GPER vasorelaxation. 

## 2. Results

### 2.1. GPER mRNA Levels Change following Gestational Age in Uterine Arteries

In uterine arteries, mRNA levels of GPER were significantly higher in P7 rats, and this increase was even greater in P14 compared with NP rats (*p* < 0.05 P7 and P14 vs. NP; Figure 1A). A trend in the increase of GPER expression was assessed by qPCR in uterine arteries following pregnancy progression (*p*= 0.06, P7 vs. P14; Figure 1A). In contrast, in mesenteric arteries, there were no differences in the mRNA levels of GPER between pregnant and non-pregnant rats or throughout gestation (Figure 1B).

### 2.2. Pregnancy Modulates Vascular Reactivity Response to G1 in Uterine but Not in Mesenteric Arteries

To provide further insights into the role of GPER, we then ascertained that the GPER agonist, G1, induces vasorelaxation of uterine (Figure 2A) and mesenteric (Figure 2B) arteries in a dose-dependent manner in both pregnant and non-pregnant rats. Notably, in uterine arteries we found a significant effect of pregnancy on G1-mediated vasorelaxation; it was modulated by gestational age (Figure 2A). G1-vasorelaxation was significantly higher in P7 compared with NP rats, with a maximum efficacy of 82.4 ± 6.0% in P7 vs. 66.5 ± 3.7% in NP rats (Figure 2A). The effect of G1 on uterine arteries significantly increased following the progression of pregnancy, with a maximum vasorelaxation of 97.8 ± 2.5% in P14 rats (Figure 2A) at 10^−6^ M concentration of G1 (Figure 2A). In contrast, in mesenteric arteries, dose-responses to G1 were similar between pregnant and non-pregnant rats and throughout gestation (Figure 2B). 

### 2.3. G1-Induced Vasorelaxation of Uterine Arteries Involves L-Type Calcium Channels 

We have previously demonstrated that G1 induces GPER-mediated vasorelaxation via the NO-cGMP pathway in uterine arteries of pregnant rats [27]. To provide further data we investigated uterine artery G1-vasorelaxation in the presence of the inhibitors of L-type Ca^2+^ channels (verapamil) and of ERK1/2 (PD 098,059). Using verapamil, the vasorelaxation induced by G1 was significantly reduced compared with the control group (18.5 ± 4.9% vs. 68.8 ± 8.4, *p* < 0.001; Figure 3A). Moreover, the vasorelaxation induced by G1 was also significantly blunted in the presence of PD 098,059 (48.0 ± 5.1% vs. 71.7 ± 5.2%, *p* < 0.05; Figure 3B). 

## 3. Discussion

The present study demonstrates that (1) expression of GPER is increased, (2) vasorelaxation to G1 is enhanced as pregnancy progresses; GPER-mediated vasorelaxation occurs through (3) L-type Ca^2+^ channels and (4) ERK1/2 pathway in uterine arteries of pregnant rats; (5) functional expression of GPER does not change through gestation in mesenteric arteries. Activation of GPER has been shown to cause vasorelaxation in rat aorta, carotid, mesenteric, renal, cerebral, and coronary arteries [30,31,32,33,34]. In addition, sex-related controversial effects and stark differences between vascular beds have also been reported [32,35,36,37,38]. Although considerable effort has been made to appreciate the role of GPER in the vasculature in both human and animal models (Appendix A), more studies are needed to fully delineate the physiological importance of GPER.

GPER together with the cognate receptors, ERα and ERβ, mediates physiological functions of estrogens, including the crucial role of regulating vascular function [25,26]. The agonist–receptor-coupling with the consequent production and release of NO and other endothelium-derived factors, determine the degree to which extent estrogens may elicit vasorelaxation [39,40]. In the present investigation, pregnancy significantly increased GPER-mediated vasorelaxation in uterine arteries, and this effect was modulated by gestational age. 

To elucidate whether the modulating role of pregnancy on GPER-mediated vasorelaxation may be vascular-bed specific, we applied G1 in a cumulative fashion on arteries obtained from the mesenteric circulation. In agreement with other studies [30,34], G1 elicited a vasorelaxing effect in mesenteric arteries. Contrary to our findings in the uterine vasculature, the similarity of responses to G1 (41–52% maximum vasorelaxation) in mesenteric arteries of both -NP and P rats, as well as at different gestational periods, suggests that pregnancy-induced modulation of GPER vascular function is specific of the uterine circulation. 

It may be implied that pregnancy-associated changes of the uterine reactivity to G1, could predict an up-regulation of GPER expression in this vascular bed. In fact, evaluation of GPER expression revealed that pregnancy modulates mRNA levels of GPER in uterine arteries, and a trend towards increase in GPER expression was observed as pregnancy progressed. A limitation of this study is that we investigated only mRNA levels of GPER. Further additional data in terms of protein expression are needed to fully support our conclusion. However, we may speculate that an increase in mRNA corresponds with an increase in the relative protein expression of GPER. Our results demonstrated a significant increase in GPER mRNA levels already after 7 days of pregnancy, with GPER expression exceeding a one-fold increase in P14 compared with NP rats. Contrariwise, we found no changes in GPER expression in mesenteric arteries, which could explain our functional results, and suggests again that pregnancy-induced modulation of GPER is specific to uterine arteries. 

The lack of modulation in mesenteric arteries may be attributed to the different adaptation of the systemic circulation to pregnancy, in terms of both reactivity and remodelling [2], with the reproductive system undertaking a greater role in both maintaining vascular adaptation [5,24] and regulating uterine blood flow throughout gestation [22]. It is also plausible that the action of estrogens is more powerful at the uterine site than in the systemic circulation through the upregulation of GPER in uterine arteries, which in turn makes this vasculature more sensitive to estrogens. An overexpression of GPER, due to increased sensitivity to estrogens, has been observed in the human melanoma tissues [41].

Although we have previously demonstrated that GPER activation elicits endothelium-dependent vasorelaxation via the NO-cGMP pathway in rat uterine arteries [27], our further elucidation of the functional signalling identified a possible smooth muscle-related mechanism involved in the vascular responses. In our current study, inhibition of L-type Ca^2+^ channels with verapamil caused about a three-times reduction of the vasorelaxation, thus suggesting that L-type Ca^2+^ channels participate in the GPER-mediated vasorelaxing effect. This conclusion is supported by other studies that have provided evidence that GPER mediates rapid cell signalling via stimulation of intracellular Ca^2+^ mobilisation [26,42,43,44,45]. It has been demonstrated that extracellular application of G1 on human aortic vascular smooth muscle cells (VSMCs) increases intracellular Ca^2+^ concentrations, and this effect is attenuated by GPER silencing [43]. Pre-incubation with the L-type Ca^2+^ channel agonist Bay K8644, enhanced the vasodilatory estrogenic effects on placental arteries [17]. Furthermore, infusion of estradiol increased perfusion pressure and coronary resistance through a non-genomic molecular mechanism involving activation of the L-type Ca^2+^ channels in rats [46]. Additionally, inhibition of L-type Ca^2+^ channels with nifedipine significantly reduced G1-induced intracellular Ca^2+^ increase in myometrial cells [47], and activation of GPER inhibited the endothelin-1-stimulated increase of intracellular Ca^2+^ concentrations in VSMCs of murine carotid arteries [45]. Although there is heterogeneity amongst cells on the mechanism through which GPER induces Ca^2+^ mobilisation [47], both the uptake of extracellular Ca^2+^ and the release of the ion from intracellular Ca^2+^ stores in cells may be involved in the GPER molecular signalling.

Studies conducted mainly in cell lines have shown that either estrogens or the GPER-selective agonist G1, can activate a number of protein kinases, including ERK1/2, via a non-ER dependent mechanism that requires increased intracellular concentrations of Ca^2+^ [48,49,50,51,52] through the L-type Ca^2+^ channels [51]. It has been demonstrated that ERK 1/2 is a signalling component of the GPER cascade that induces NO production in human endothelial cells [53], with the same mechanism [54] through which estrogen-mediated rapid intracellular signalling regulates NO synthesis [55]. 

Therefore, once the participation of the L-type Ca^2+^ channels was established, we sought to investigate the involvement of ERK 1/2 signalling in the vascular effects elicited by GPER in uterine arteries. 

Incubation with the ERK 1/2 inhibitor, PD 098059, showed that vasorelaxation in response to G1 was attenuated by 24%, which is suggestive of a partial contribution of ERK1/2 in the mechanism of action of GPER in uterine arteries. The partial inhibition suggests that other protein kinases may be involved in the signalling cascade activated by GPER and warrant further investigation.

Consistent with our data, other studies provided evidence that GPER promotes activation of ERK1/2 in cancer cell lines [50,56], in rat heart [57,58], and in human umbilical vein smooth muscle cells [43].

In conclusion, we have shown that activation of GPER induces vasorelaxation of reproductive (uterine) and systemic (mesenteric) arteries. In uterine arteries, GPER-mediated vasorelaxation increases as pregnancy progresses. This can be explained by an upregulation of GPER gene expression in uterine arteries during pregnancy, which may be of importance in regulating hemodynamic changes of pregnancy in the reproductive system. Moreover, the emerging contribution of L-type Ca^2+^ channels and ERK 1/2 signalling provides a new perspective for understanding the vascular mechanisms involved in GPER-mediated vasorelaxation of uterine arteries in pregnant rats. The modulating effect of both GPER expression and GPER vasorelaxation, associated with gestational age, is supportive of a physiological role for GPER in the uterine vascular adaptation and may offer a novel therapeutic target through which to selectively improve local uterine circulation in pregnancies with compromised uteroplacental blood flow. 

## 4. Methods 

### 4.1. Experimental Animals 

The present study was performed in accordance with the European Guidelines on the protection of animals used for scientific purposes (Directive 2010/63/EU). Arteries were isolated from animals used in studies approved by the local ethical committee (OPBA) at the University of Calabria and the Italian Ministry of Health (Ufficio VI, n. 295/2016-PR, March 2016 and n. 530/2021-PR, July 2021).

Sprague–Dawley rats were housed under controlled conditions on a 12 h light/dark cycle at 20–22 °C; commercial chow and tap water were provided *ad libitum*. Experiments were performed on age-matched (12–15 weeks old) non-pregnant (NP; *n* = 8) and pregnant rats at gestational days 7 (P7; *n*= 8) and 14 (P14; *n*= 8). Pregnant rats were obtained by mating a female in proestrus with a fertile male overnight and checked the following morning. Detection of spermatozoa in the vaginal smear was used to confirm day 1 of pregnancy. Rats were euthanized with isoflurane, followed by decapitation with a small animal guillotine. Rapidly, the uterus and mesentery were both removed and collected in ice-cold HEPES-physiological saline solution (HEPES-PSS, in mmol/L: sodium chloride 141.8, potassium chloride 4.7, magnesium sulfate 1.7, calcium chloride 2.8, potassium phosphate 1.2, HEPES 10.0, EDTA 0.5, and dextrose 5.0).

### 4.2. Pressure Myography

Radial uterine and third-order mesenteric arteries (diameter < 300 µm) obtained from NP, P7, and P14 rats were dissected free from perivascular connective and adipose tissue in ice-cold HEPES-PSS. Arterial segments (1–2 mm long) were transferred to the chamber of a small-vessel pressure myograph. One end of the artery was tied onto a glass cannula, any luminal content was flushed off by increasing intraluminal pressure before securing the distal end onto a second glass cannula using a servo-null pressure system (Living Systems Instrumentation, St. Albans City, VT, USA). All arteries were continuously superfused with HEPES-PSS at 37 °C, pressurised to 50 mmHg, and equilibrated for 45 min under no-flow conditions before the experiment started. Lumen diameter was measured by trans-illuminating each arterial segment in conjunction with data-acquisition software (Ionoptix, Westwood, MA, USA) to continuously record lumen diameter. Following equilibration, arteries were pre-constricted with phenylephrine (10^−7^) to produce a 40–50% reduction in baseline diameter [59]. Once pre-constriction was achieved and remained stable for about 10 min, a concentration–response curve to the specific agonist of GPER, 1-(4-(-6-Bromobenzol(1,3)diodo5-yl)3a,4,5,9b-tetrahidro-3Hcyclopenta(c-)ethenone-8yl)ethenone (G1; 10^−10^–10^−6^ M), was obtained (a representative pressure myography trace is reported in Appendix A). To investigate the contribution of L-type calcium channels and ERK1/2 signalling, uterine arteries were pre-incubated with the inhibitors verapamil (10^−5^ M) or PD 098059 (10^−6^ M), respectively, for 30 min before pre-constriction with phenylephrine (10^−6^ M) to produce a 40–50% reduction in baseline diameter, and vasorelaxation to G1 was assessed. HEPES-PSS without Ca^2+^ plus papaverine (10^−4^ M) was added to calculate the maximum vessel diameter at the end of each experiment.

### 4.3. Total RNA Extraction and Quantitative Real-Time Polymerase Chain Reaction (qPCR) 

Uterine and mesenteric arteries were isolated as described above and preserved at −80 °C. Frozen samples were homogenised with a motor-driven homogeniser, and total RNA was isolated using Trizol reagent (Life Technologies, Milan, Italy), according to the manufacturer’s instructions. RNA was quantified spectrophotometrically, and quality was checked by electrophoresis, through agarose gels stained with ethidium bromide. Total cDNA was synthesised from RNA by reverse transcription using the murine leukemia virus reverse transcriptase (Life Technologies, Milan, Italy), following the protocol provided by the manufacturer. The expression of GPER was quantified by real-time qPCR using the Step One^TM^ sequence detection system (Applied Biosystems Inc., Milan, Italy). Specific primers for GAPDH and GPER were designed using Primer Express version 2.0 software (Applied Biosystems). The following sets of primers were used: for GAPDH (internal control), 5′-CAAGGCTGTGGGCAAGGT-3′ (forward), 5′-GGAAGGCCATGCCAGTGA-3′ (reverse); for GPER, 5′-CCTGGACGAGCAGTATTACGATAT-3′ (forward), 5′-CGCTGCTGTACATGTTGATCTG-3′ (reverse). Assays were performed in triplicate.

### 4.4. Drugs and Chemicals 

G1 and PD098059 were purchased from TOCRIS (R&D Systems, Milan, Italy); Verapamil was purchased from Santa Cruz (Heidelberg, Germany). All the other chemicals were purchased from Sigma-Aldrich (Milan, Italy). 

G1 stock solution was dissolved in DMSO, and small aliquots were frozen prior to use. All the other drugs tested were prepared daily. Phenylephrine and papaverine were dissolved in HEPES-PSS; verapamil and PD 098059 inhibitors were dissolved in DMSO. All drugs were kept on ice, in lightproof vials, and further diluted in HEPES-PSS as required.

### 4.5. Statistical Analysis 

Vasorelaxation to G1 was expressed as percentage of maximally vasorelaxed diameter. Data are expressed as mean ± SEM, and *n* is the number of rats used. Differences between groups were determined by one-way ANOVA (for relative mRNA levels), and two-way ANOVA for repeated measures analysis, followed by Tukey’s post hoc test where appropriate (for vasorelaxant responses). Differences in vasorelaxation following incubation with inhibitors were determined using paired Student’s *t*-test, performed by Prism 8 software (GraphPad Software Inc., La Jolla, CA, USA). Differences were considered significant at *p* < 0.05.

## Figures and Tables

**Figure 1 ijms-23-05996-f001:**
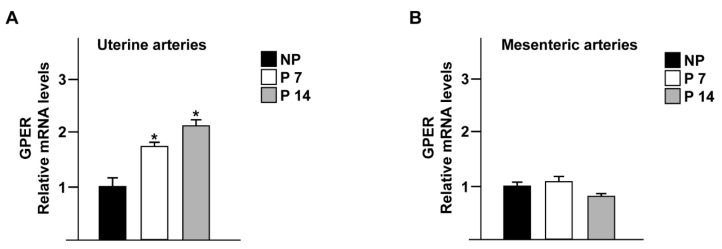
Pregnancy increases GPER expression in uterine but not in mesenteric arteries. GPER expression was quantified by qPCR in uterine (**A**) and mesenteric (**B**) arteries from non-pregnant (NP) and pregnant rats at 7 (P7) and 14 (P14) days of gestation. GPER mRNA was significantly higher in uterine arteries from both P7 and P14 rats compared with NP rats (**A**). Expression levels were not affected by pregnancy in mesenteric arteries (**B**). * *p* < 0.05, P7, P14 vs. NP, by one-way ANOVA. *n* = 3 independent experiments per group performed in triplicate.

**Figure 2 ijms-23-05996-f002:**
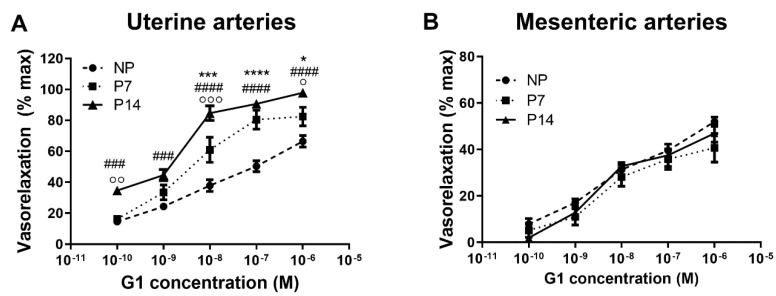
Pregnancy modulates concentration–responses to G1 in uterine arteries. In uterine arteries (**A**) of pregnant rats, vasorelaxation to G1 was significantly higher at 7 days (P7, *n* = 8) compared with non-pregnant (NP, *n* = 8) rats, and this response was even greater at 14 days of pregnancy (P14, *n* = 8). In mesenteric arteries (**B**), pregnancy had no effect on responses to G1 (NP, *n* = 6; P7, *n* = 5; P14, *n* = 6). P7 vs. NP: * *p* < 0.05, *** *p* < 0.001, **** *p* < 0.0001; P14 vs. NP: ### *p* < 0.001, #### *p* < 0.0001; P7 vs. P14: ° *p* < 0.05, °° *p* < 0.01, °°° *p* < 0.001, by two-way ANOVA.

**Figure 3 ijms-23-05996-f003:**
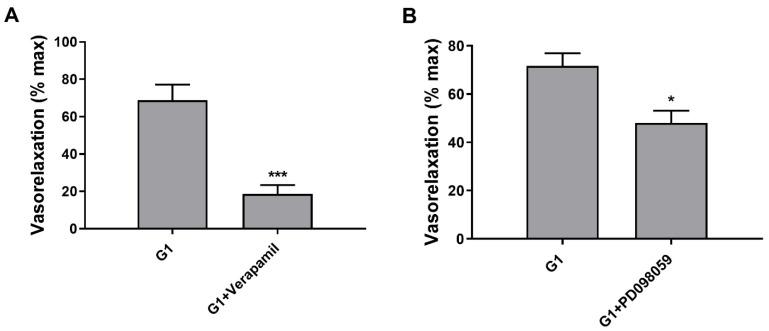
L-type Ca^2+^ channels and ERK1/2 signalling contribute to G1-mediated vasorelaxation. In uterine arteries, inhibition of L-type calcium channels with verapamil (10^−5^ M, (**A**)) and of ERK1/2 pathway with PD 098,059 (10^−6^ M, (**B**)) significantly reduced vasorelaxation to G1 (10^−8^ M). *** *p* < 0.001, G1 vs. G1 + verapamil (*n* = 6); * *p* < 0.05, G1 vs. G1 + PD 090,859 (*n* = 5), by *t*-test.

## Data Availability

Data available on request from the corresponding author.

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
