# Peer review of "G-Protein-Coupled Estrogen Receptor Expression in Rat Uterine Artery Is Increased by Pregnancy and Induces Dilation in a Ca2+ and ERK1/2 Dependent Manner"

_ijms, 2022, doi:10.3390/ijms23115996_

Round 1

Reviewer 1 Report

Dear Editor/Author

The ms entitled " G-protein estrogen receptor expression....... ERK1/2 dependent manner" described the role of GPER in mediating the vasorelaxation of uterine arteries and in regulating uteroplacental blood flow. The concept of the research is reasonable but the following point may be considered:

Minor remarks:

  1. Title could be modulated to include G-protein-coupled estrogen receptor.
  2. English could be improved: P2 line 60; P2 line 62; P2 line 63; P3 line 104; P4 line 123; P4 line 125; P5 line 176; P6 line213 spelling
  3. There is some redundancy: lines125-128; and line 135.

Major Remarks:

  1. Methodology: I) The concentration of phenylephrine that gives 40-50% reduction in baseline diameter should be mentioned, II)Verapamil 10-5M: Doesn't that concentration reduce the action of Phenylephrine? This may affect the comparisons, Calrify, please: HEPES-PSS without Ca2+ plus papaverine was added to calculate..... Does this mean that you changed the buffer that contained phenylephrine? Have you added phenylephrine again?, Does 10-4 M papaverine cause maximum relaxation?! IV) Looking for GPER mRNA expression is not sufficient. To verify your data you should have done Western plotting for GPER, V) To investigate the role of ERK1/2 in vasorelaxation, the phosphorylation of ERK1/2 should be examined, VI) Statistics: Why 2-way ANOVA, not One-way ANOVA? Which figures were analyzed by 2-way ANOVA then by Tukey's? What kind of Student's t-test was used? paired? non-paired? please specify
  2. Results: Fig. 1 What is N for NP in part B?; Fig. 2 please specify the type of statistics used to ompare the 3 curves in part A?; Figure 3: there is no mention in the legend to part B, and mention of the type of statistics. Is it t-test?
  3. Discussion: P4 lines 133-135 Is that due to estrogen or to GPER? ince there is no added estrogen in your experiment; P5 lines151-153 Would one-fold increase as shown in Fig. 1 be sufficient to report significant increase in GPER expression. There should be a higher cut-score. That is probably why Western blot is required; P5 lines 159-161 How could estrogen levels be (more powerful) at the uterine site of action than those in the systemic circulation?! Are you suggesting variable regional distribution of estrogen? Unlikely. Could it be GPER differences?; P5 lines 170-173: please see my comments on methods about the effect of verapamil; Table 1 extending over pages 6-13 is superfluous! This is not a review manuscript. You could cite one or 2 examples to clarify your point concerning the role of GPER in the vasculature. Table should be removed.
  4. References: List is extensive: Should be shortened by 50% or more; None of the references has been written to the Journal style or to the acceptable standards of citation (i.e. Last name first then initials); Many references need clean -up of unnecessary information; Reference 29 is misplaced. 

Author Response

Minor remarks:

  1. Title could be modulated to include G-protein-coupled estrogen receptor.

Added in the title

  1. English could be improved: P2 line 60; P2 line 62; P2 line 63; P3 line 104; P4 line 123; P4 line 125; P5 line 176; P6 line213 spelling

Thanks for the suggestion we revised it

  1. There is some redundancy: lines125-128; and line 135.

Thank you, we revised the text

Major Remarks:

  1. Methodology: I) The concentration of phenylephrine that gives 40-50% reduction in baseline diameter should be mentioned,

It was added in P 4.2. line 274.

  1. II) Verapamil 10-5M: Doesn't that concentration reduce the action of Phenylephrine? This may affect the comparisons,

To have comparable conditions, both control arteries and Verapamil treated arteries were contracted with phenylephrine always at 40-50% of the original diameter. In treated arteries the presence of Verapamil required a somewhat higher concentration of phenylephrine than in the control. It was added in P 4.2. line 282

Calrify, please: HEPES-PSS without Ca2+ plus papaverine was added to calculate..... Does this mean that you changed the buffer that contained phenylephrine? Have you added phenylephrine again?, Does 10-4 M papaverine cause maximum relaxation?!

Calcium-free Hepes-PSS solution with papaverine was used at the end of each experiment in order to obtain the maximum vessel diameter (we clarify it in P 4.2. line 284) in order to normalize GPER-vasodilation always to the maximum diameter of each vessel. In this regard, a new solution of HEPES-PSS without calcium with the addition of papaverine and in the absence of Phe is used. The maximum diameter obtained usually coincides with the original diameter of the vessel at equilibrium before it is contracted with the Phe suggesting the absence of vascular tone.

  1. IV) Looking for GPER mRNA expression is not sufficient. To verify your data you should have done Western plotting for GPER.

We agree that western blotting would have supported our conclusion of the GPER increase during pregnancy. However, since an increase in mRNA usually corresponds to an increase in the relative protein, it is plausible that this also happens for GPER. This was added in the discussion, line 158-161.

  1. V) To investigate the role of ERK1/2 in vasorelaxation, the phosphorylation of ERK1/2 should be examined,

We have determined the involvement of ERK1 / 2 in GPER-relaxation through the use of a specific inhibitor of ERK1 / 2 (pharmacological approach) often used in studies that aim to highlight the transduction pathway underlying a given physiological process. . We agree that further support confirming ERK 1/2 involvement may come from the presence of ERK 1/2 phosphorylation but this does not preclude our result.

  1. VI) Statistics: Why 2-way ANOVA, not One-way ANOVA? Which figures were analyzed by 2-way ANOVA then by Tukey's? What kind of Student's t-test was used? paired? non-paired? please specify

We apologise for the lack of clarity, and we have now included this information in figure legends, Statistical analysis and amended graph A in Fig.1 accordingly.

  1. Results: Fig. 1 What is N for NP in part B?; Fig. 2 please specify the type of statistics used to ompare the 3 curves in part A?; Figure 3: there is no mention in the legend to part B, and mention of the type of statistics. Is it t-test?

We apologise for missing this information, which is now stated in the respective figure legends and Statistical analysis.

  1. Discussion: P4 lines 133-135 Is that due to estrogen or to GPER? ince there is no added estrogen in your experiment;

We are sorry for not having clearly stated that the effect observed in our study is due to an increase in the expression of GPER which we have now reported in the revised text, line 137-146

P5 lines151-153 Would one-fold increase as shown in Fig. 1 be sufficient to report significant increase in GPER expression. There should be a higher cut-score. That is probably why Western blot is required;

Our results suggest a significant increase of GPER- vasodilation in association with the increase of GPER  mRNA by pregnancy. It has been reported that even lower amount of mRNA is sufficient to maintain protein levels (doi: 10.1093/molehr/gaq043).

 P5 lines 159-161 How could estrogen levels be (more powerful) at the uterine site of action than those in the systemic circulation?! Are you suggesting variable regional distribution of estrogen? Unlikely. Could it be GPER differences?;

Estrogens can be more powerful at the uterine site than in the systemic circulation throught an upregulation of GPER in the uterine arteries ( we reported it in the revised text,  line  171-178)

P5 lines 170-173: please see my comments on methods about the effect of verapamil;

The reduction of GPER-vasorelaxation in the presence of verapamil, a specific inhibitor of the L-Type calcium channels, suggests the involvement of this channel in the action of the GPER.

Table 1 extending over pages 6-13 is superfluous! This is not a review manuscript. You could cite one or 2 examples to clarify your point concerning the role of GPER in the vasculature. Table should be removed.

We agree with the reviewer that Table 1 disrupts the flow of the manuscript. Therefore, we have included the table as Supplementary Materials.

  1. References: List is extensive: Should be shortened by 50% or more; None of the references has been written to the Journal style or to the acceptable standards of citation (i.e. Last name first then initials); Many references need clean -up of unnecessary information; Reference 29 is misplaced. 

The list of reference has now been shortened and the style has been amended as per Journal style requirements

Reviewer 2 Report

This manuscript is a follow up of a previous work published in Plos One where the authors show that the GPER induces vasodilation in rat uterine arteries via the NO. Here, they demonstrated that this vasodilation is i) specific of uterine arteries as opposed to mesenteric arteries, and ii) induced via a Ca2+ and Erk1/2 dependent manner.

  • Can the authors mention the expression levels of Esr1, Esr2, and GPER mRNA in uterine arteries in the different conditions (pregnant and non pregnant) and not only GPER mRNA. Express on absolute numbers of mRNA and not at relative levels to indicate what is the major abundant gene?

  • The authors will need to show some representative images of the dilatation in response to the concentration response to DMSO or G1 conditions?

  • Is G1-induced relaxation dependent on the production of NO? Show data or discuss previous published data.

  • Calcium entry inhibition and ERK1/2 blockade are also known to block or at least reduce contractility. The authors should show the level of precontraction obtained before G1-induced relaxation.

  • Is the effect selective of GPER or is this a global increase in the relaxing capacity of the uterine artery? For example, is acetylcholine-mediated relaxation also increased in pregnant rats compared to NP rats?

  • Discussion: is it possible that GPER becomes overexpressed in situations requiring a stronger response of the estrogen-dependent pathway as this is the case in pregnancy? This point could be discussed in comparison of the other situations where GPER seems to be involved.

  • Table 1: Better organize the results in Table 1- in other words, cluster the different published works according to the diverse species (human, rats, mice ..) and performed methods as a follow-up.

  • In the introduction, better cite reviews or other original papers from other laboratories, and not only the authors’s work regarding the effects of estrogens on vascular functions and rapid vasodilation.

Author Response

  • Can the authors mention the expression levels of Esr1, Esr2, and GPER mRNA in uterine arteries in the different conditions (pregnant and non pregnant) and not only GPER mRNA. Express on absolute numbers of mRNA and not at relative levels to indicate what is the major abundant gene?

 Expression of Esr1 and Esr2 in the uterine circulation in the non pregnant and pregnant state has been extensively studied [as reviewed in doi: 10.3390/ijms21124349 (ref 16)] unlike the influence of pregnancy on GPER. The purpose of our study is not to compare GPER with Ers1 e Ers2, but rather demonstrate the role of GPER in the uterine vasculature during pregnancy. 

  • The authors will need to show some representative images of the dilatation in response to the concentration response to DMSO or G1 conditions?

Thanks for the suggestion, we have included  representative  traces  as Supplementary Materials.

  • Is G1-induced relaxation dependent on the production of NO? Show data or discuss previous published data.

Yes, it is as we showed in our previous study  ( doi: 10.1371/journal.pone.0141997. eCollection 2015) and reported in this one in line 180 & ref 26.

  • Calcium entry inhibition and ERK1/2 blockade are also known to block or at least reduce contractility. The authors should show the level of precontraction obtained before G1-induced relaxation.

To have comparable conditions, both control arteries and treated arteries with the Calcium or ERK1/2inhibitors were contracted with phenylephrine always at 40-50% of the original diameter. In treated arteries the presence of the inhibitor required a somewhat higher concentration of phenylephrine than in the control. It was added in methods line 274 and 282

  • Is the effect selective of GPER or is this a global increase in the relaxing capacity of the uterine artery? For example, is acetylcholine-mediated relaxation also increased in pregnant rats compared to NP rats?

Since in this study we investigated the uterine artery vasodilation induced by G1, a specific agonist of GPER whose expression increases during pregnancy, we believe that the increase of G1-vasodilation  during pregnancy is due to GPER.

Furthermore, Acetylcholine-vasodilation of the uterine arteries is reduced in pregnancy compared to non-pregnancy. From our experience on rat uterine arteries in pregnancy, the maximum vasodilation induced by Ach at 10-5M is about 40% against 60% in non-pregnant at the same concentration of Ach.

  • Discussion: is it possible that GPER becomes overexpressed in situations requiring a stronger response of the estrogen-dependent pathway as this is the case in pregnancy? This point could be discussed in comparison of the other situations where GPER seems to be involved.

 Thanks for the suggestion, this concept is now added in discussion, line 169-178

  • Table 1: Better organize the results in Table 1- in other words, cluster the different published works according to the diverse species (human, rats, mice ..) and performed methods as a follow-up.

We thank the reviewer for this suggestion and we have amended the table accordingly (now included as Supplementary Materials).

  • In the introduction, better cite reviews or other original papers from other laboratories, and not only the authors’s work regarding the effects of estrogens on vascular functions and rapid vasodilation.

References 12-25 show the work of various experts in the field. I'm sorry if I forgot someone, however I added another ref.25 (doi: 10.3390/ijms21124349) relate to the role of estrogen in the uterine vasculature.

Round 2

Reviewer 1 Report

Minor editing comments:

1. Line 22 "the was..." change to "there was...."

2. Line 177-178 "which is estrogen sensitivity" change to " which is due to estrogen sensitivity"

3. Line 275 Reference 61 probably should be 60. Please check.

Author Response

Dear Reviewer,

 Thanks for the suggestions

  1. Line 22 "the was..." change to "there was...."

done

  1. Line 177-178 "which is estrogen sensitivity" change to " which is due to estrogen sensitivity"

done

  1. Line 275 Reference 61 probably should be 60. Please check.

You are right it was corrected, thanks.